# Sleep Architecture and Sleep-Related Breathing Disorders of Seafarers on Board Merchant Ships: A Polysomnographic Pilot Field Study on the High Seas

**DOI:** 10.3390/ijerph20043168

**Published:** 2023-02-10

**Authors:** Fiona Kerkamm, Dorothee Dengler, Matthias Eichler, Danuta Materzok-Köppen, Lukas Belz, Felix Alexander Neumann, Birgit-Christiane Zyriax, Volker Harth, Marcus Oldenburg

**Affiliations:** 1Institute for Occupational and Maritime Medicine Hamburg (ZfAM), University Medical Center Hamburg-Eppendorf (UKE), 20459 Hamburg, Germany; 2Preventive Medicine and Nutrition, Institute for Health Services Research in Dermatology and Nursing (IVDP), University Medical Center Hamburg-Eppendorf (UKE), 20459 Hamburg, Germany

**Keywords:** ESS, occupational medicine, OSA, obstructive sleep apnea, polysomnography, pupillometry, seafaring, sleep, sleepiness

## Abstract

As seafarers are assumed to have an increased risk profile for sleep-related breathing disorders, this cross-sectional observational study measured (a) the feasibility and quality of polysomnography (PSG) on board merchant ships, (b) sleep macro- and microarchitecture, (c) sleep-related breathing disorders, such as obstructive sleep apnea (OSA), using the apnea–hypopnea index (AHI), and (d) subjective and objective sleepiness using the Epworth Sleepiness Scale (ESS) and pupillometry. Measurements were carried out on two container ships and a bulk carrier. A total of 19 out of 73 male seafarers participated. The PSG’s signal qualities and impedances were comparable to those in a sleep laboratory without unusual artifacts. Compared to the normal population, seafarers had a lower total sleep time, a shift of deep sleep phases in favor of light sleep phases as well as an increased arousal index. Additionally, 73.7% of the seafarers were diagnosed with at least mild OSA (AHI ≥ 5) and 15.8% with severe OSA (AHI ≥ 30). In general, seafarers slept in the supine position with a remarkable frequency of breathing cessations. A total of 61.1% of the seafarers had increased subjective daytime sleepiness (ESS > 5). Pupillometry results for objective sleepiness revealed a mean relative pupillary unrest index (rPUI) of 1.2 (SD 0.7) in both occupational groups. In addition, significantly poorer objective sleep quality was found among the watchkeepers. A need for action with regard to poor sleep quality and daytime sleepiness of seafarers on board is indicated. A slightly increased prevalence of OSA among seafarers is likely.

## 1. Introduction

It is well known that seafarers are exposed to extreme working and living conditions on board, which can also have an impact on sleep patterns and daytime sleepiness. In addition to considerable psychophysical stress due to time pressure and long, irregular working hours, as well as noise, vibration, and strong ship movements can represent further stress factors [1]. Matsangas and Shattuck [2] found that 91.6% of seafarers had their sleep disturbed by at least one environmental factor (noise, temperature, light, ship movements, odors, poor bedding conditions).

Moreover, shift work—which is usually in a six-on, six-off or four-on, eight-off cycle on board—hardly allows enough time for recovery phases [3]. In line with this, various studies have shown that while sleep duration on board is generally shortened and sleepiness increased, the watchkeepers were usually more affected than the day workers. Although Oldenburg and Jensen [4] observed that day workers also had alarmingly reduced sleep duration (TST 5.8 h) and sleep efficiency (SE% 72.7%), the watchkeepers reached significantly worse values (TST 5.5 h; SE% 67.9%). Similar observations were made by Shattuck and Matsangas [5]. They found that watchkeepers slept worse (TST < 7 h: 67.6% of watchkeepers and 48.3% of day workers) and reported more excessive daytime sleepiness (ESS > 10: 45.5% of watchkeepers and 30.9% of day workers). Likewise, differences in sleep duration between nautical officers (4.7 h), crew ratings on deck (5 h), and engine room personnel (5.4 h) were detected actigraphically. In addition, the officers also subjectively reported an existing sleep deficit more frequently [6]. In general, the World Maritime University 2020 report also confirmed that seafarers were unable to comply with legal work/rest hours due to both inadequate staffing levels and strict labor contracts, putting them under additional pressure, which could lead to overwork and fatigue [7].

Furthermore, seafarers on two-section watch systems (e.g., six-on, six-off system) showed extremely short meal-to-sleep intervals and consumed food during the circadian night, which was both associated with greater weight gain and a higher percentage of body fat [8].

Based on these findings, a variety of subjective and objective measurement methods for fatigue, sleepiness, and sleep behavior have already been used on board ships. An overview of these previously used measurement methods on board was compiled in a review [9]. It was found that questionnaires were frequently used, but these can easily be influenced by recall bias or social desirability. The most common method of objective sleep measurement so far has been actigraphy. This is particularly suitable for investigating sleep behavior over a longer period of time. However, a more detailed representation of the sleep architecture or the diagnosis of sleep-related breathing disorders is not possible with actigraphic movement measurements. As part of the interdisciplinary project “e-healthy ship”, which aims to optimize health management on board ships operating without doctors, ambulatory polysomnography (PSG)—the gold standard of sleep diagnostics [10]—was therefore used on merchant ships for the first time to our knowledge.

Previous comparable polysomnographic measurements in a maritime context have only been performed in a bridge simulator to investigate daytime sleepiness during a shift [11]. This revealed a strikingly increased sleepiness in the form of a tendency to fall asleep, especially during the shift from midnight to 4 a.m.

A polysomnographic sleep assessment of seafarers during their normal working day on board has not yet been conducted. However, studies exist that have used polysomnography to examine sleep patterns in similarly mobile work contexts, such as train drivers [12] or aircrews [13]. As no disturbances of the ambulatory PSG measurements (e.g., due to vibrations or turbulences) were mentioned in these study settings, a good feasibility of the ambulatory PSG could also be expected in a maritime working environment. However, in addition to the feasibility of PSG on board, it should also be investigated whether ship movements or vibrations from merchant ships are disturbing factors for the PSG measurements on the high seas, as these could lead to a more difficult evaluation or could be falsely interpreted as participants’ movements.

In addition to the detailed recording of sleep architecture at sea, the ability to diagnose sleep-related breathing disorders is a major advantage of PSG over the other methods used to date to measure sleep or sleepiness on board ships. Obstructive sleep apnea (OSA) is a breathing disorder caused by functional instability of the upper airway during sleep. The collapse of the upper airway leads to arousals and waking reactions associated with apneas and hypopneas, as well as reduced oxygen saturation of the blood. The most common symptom is increased daytime sleepiness [10], which is generally a major safety risk on board. An investigation of accidents at sea found a sleep connection with the incidents in 86% of the cases analyzed. In the majority of these reports (34%), accidents were related to “sleep loss as a way of life” (waking at unusual times, daytime sleep, working instead of sleeping, and sleep hygiene) [14]. Furthermore, multiple reviews support the association of occupational fatigue with errors, accidents, and injury [15,16,17]. In addition, cardiovascular diseases and metabolic disorders are also associated with OSA [10]. This aspect is particularly relevant, as an increased risk of cardiovascular disease in seafarers has been repeatedly reported [18,19,20,21,22].

In general, it is assumed that OSA is widespread in the population [23]. To our knowledge, an investigation of the prevalence of OSA for the occupational collective of seafarers has not been conducted before.

The aim of this field study was to assess the feasibility and quality of polysomnography on board merchant ships as well as to gain a deeper insight into the sleep architecture and possible sleep-related breathing disorders of seafarers.

## 2. Materials and Methods

### 2.1. Study Population

As part of the interdisciplinary project “e-healthy ship”, doctors and researchers accompanied two container ships and a bulk carrier and examined the crew members on board within the framework of a pilot study. Nineteen out of 73 male seafarers participated in this part of the cross-sectional field study (participation rate: 26.0%). All seafarers were invited to participate in this study (no exclusion criterion).

The participants could be divided into officers, who have completed three years of training at a maritime university and were responsible for management on board, and non-officers, who attended basic courses at maritime schools over a three-year period. Furthermore, one main focus of this study was the comparison of watchkeepers (majority of officers and non-officers on deck, who often worked 24-h shifts due to mandatory navigation maneuvers) and day workers (engine room personnel, electricians, and galley personnel, who could often maintain a regular workday of 8 h).

Due to the comprehensive and time-consuming examination procedure, only a limited number of polysomnographic examinations could be conducted on board, meaning that this study must have been regarded as a pilot project investigating whether the effort of further PSG measurements on board would be worthwhile with larger collectives. Although crew members who were known snorers were specifically requested when selecting the subjects, this did not constitute an inclusion criterion for this study. No further differences between participating and non-participating seafarers were observed in age, BMI, nationality, rank, smoking, and sleep problems. Participation in this study was voluntary, and the evaluation was pseudonymized. In addition, the study design was approved by the Ethics Committee of the Hamburg Medical Association (no. PV 7174).

### 2.2. Patient and Public Involvement Statement

Patients and the public were not directly involved in the design, conduct, reporting, or dissemination plans of the research. All patients were informed that the dissemination of the results would be accessible on request.

### 2.3. Methods

#### 2.3.1. Polysomnography

The sleep of the seafarers was measured by means of ambulatory polysomnography (SOMNOscreen Plus, SOMNOmedics GmbH, Randersacker, Germany) by two trained physicians once per participant. No fixed sleep time or minimum sleep duration was specified. An overview of the start of the time in bed (TIB), divided into day workers and watchkeepers, can be seen in Table 1.

The PSG recordings included six EEG electrodes (electroencephalogram: C3, C4, M1, M2, REF = central reference electrode, and GND = grounding). In addition, eye movements were recorded by electrooculography (EOG), and facial muscle tone by electromyography (EMG) of the chin. Furthermore, a one-channel electrocardiogram (ECG) was recorded, as well as an EMG on both legs over the tibialis anterior muscle. Abdominal and thoracic respiratory expansions were detected using effort sensors. Further respiratory events could be measured using a nasal cannula as a pressure sensor, and snoring events could be recorded with a microphone placed next to the larynx. Capillary oxygen saturation was monitored by light-sensitive finger-pulse oximetry. A schematic illustration of the PSG cabling is shown in Figure 1.

In addition, the participants were recorded on camera, and the ship’s vibration was measured. The latter was carried out using an external activity sensor (SEN620 from SOMNOscreen plus), which was connected to the porthole or the ship’s side with the aid of a cable extension (approx. 3 m).

The polysomnography recordings were viewed in 30-s epochs and manually analyzed according to the criteria of the American Academy of Sleep Medicine (AASM) of 2012 [24] by a trained sleep technician as well as reviewed and assessed by an experienced sleep physician. The analysis software DOMINO (version 3.0.0.4, SOMNOmedics GmbH, Randersacker, Germany) was used for this purpose.

The diagnosis of obstructive sleep apnea (OSA) was made according to the criteria of the International Classification of Sleep Disorders (ICSD-3): AHI ≥ 15 (apnea–hypopnea index) or an AHI ≥ 5 in combination with typical clinical symptoms (e.g., sleepiness, fatigue), or relevant comorbidities (e.g., hypertension, type 2 diabetes mellitus, coronary artery disease). In addition, the breathing disorder must not be explained by any other sleep disorder, medical condition, medication, or other substances [25]. An AHI < 5 is considered normal in adults, 5 to <15 as mild OSA, 15 to <30 as moderate OSA, and ≥30 as severe OSA [10].

#### 2.3.2. Pupillometry

Pupillometry is an objective method for recording daytime sleepiness in which the spontaneous and unconscious movements (oscillations) of the pupil are recorded without a light stimulus while the test person wears completely darkening pupillometry glasses in a darkened room for a measurement period of 11 min. Since pupil width is controlled exclusively by the autonomic nervous system, a high activation level of the autonomic nervous system is expressed in a stable pupil width. However, in the case of severe daytime sleepiness, however, fluctuations in pupil width, so-called “fatigue waves”, can be observed, which increase with the degree of sleepiness [26].

Nine of the 19 subjects were additionally examined for sleepiness-related effects by pupillometry (mostly before the start of the sleep period). The “Fit-For-Duty” device (AMTech Pupilknowlogy GmbH) was used for the pupillometric measurements. Further recordings were not possible due to a damaged device.

The relative pupillary unrest index (rPUI) was used as an evaluation parameter of pupillometry. If the rPUI was <1.02, it was considered “normal”; if it was ≥1.02 and <1.53, the measurement was considered “conspicuous”, and if the index was ≥1.53, the subject was classified as “unfit for duty” [4].

#### 2.3.3. Questionnaires

In addition, interviews were conducted, and questionnaires were used to collect demographic data, lifestyle parameters, and the subjective assessment of sleep problems. Furthermore, the Epworth Sleepiness Scale (ESS) was used to assess subjective daytime sleepiness [27]. This questionnaire is a widely used measurement method that examines the likelihood of falling asleep in eight typical everyday situations. A high degree of internal consistency in this questionnaire has been demonstrated (Cronbach’s alpha = 0.88) [28].

### 2.4. Statistical Analysis

Statistical analysis was carried out using SPSS (version 27, IBM Corporation, Armonk, NY, USA). The Shapiro–Wilk test was used to test for the normal distribution of the data. If variables were not normally distributed, non-parametric tests (Mann–Whitney U-test) were applied; otherwise, the *t*-test was used if the distribution was normal. For a better comparison of the collected sleep parameters of the seafarers with reference parameters from the normal male population, a one-sample *t*-test was also used in this case for non-normally distributed parameters, and the means, as well as 95% confidence intervals, were given. Fisher’s exact test was used to analyze frequencies between the parameters. In addition, correlations were made using the Pearson and Spearman tests to examine the effects of age, seafaring experience, and length of stay on board on the polysomnographic measured parameters. All reported *p*-values were two-sided, and a *p*-value < 0.05 was considered statistically significant.

## 3. Results

### 3.1. Quality of Polysomnography on Board Merchant Ships

To evaluate the feasibility and quality of polysomnography on board and on the high seas, registered artifacts (e.g., due to ship motion or vibration), as well as the impedance and signal quality of the measurements, were assessed.

The impedance reflects the skin resistance. Thus, impedance quality refers to the correct placement of the electrodes on the skin. The signal quality is, on the one hand, dependent on the impedance quality and, on the other hand, includes various artifacts (patient-related/biological as well as technical disturbances) as well as external possibly interfering parameters (e.g., ambient vibration/movements or electrosmog).

The impedances indicated by the DOMINO analysis software (version 3.0.0.4) showed good values in 57.9% of the cases and sufficient values in 42.1%. The general signal quality of the PSG measurements was rated as good in all cases (Table 2).

No global artifacts or sleep stage artifacts (artifacts of the EEG, EOG, or EMG) occurred during the PSG measurements. Pulse oximetry artifacts (O_2_-artifacts) were observed in 36.8% of subjects with a mean artifact value of 6.7% (SD 6.3%) of TIB. In addition, the artifacts occurred during respiratory analysis in 10.5% with a mean deviation of 30.5% (SD 40.1%) and in the heart rate (HR) determination in 15.8% of the measurements, whereby the HR-artifacts were negligibly small with a mean value of 0.2% of the TIB (SD 0.1%) (Table 3).

In addition, this study tested whether vibrations could be recorded with the external activity sensor SEN620 of the SOMNOscreen plus, which is conceptually used for tremor recording in Parkinson’s patients. This sensor attributed to potentially ship-induced disturbances in the PSG evaluation or sleep quality of the subjects. However, despite subjectively perceptible ship vibrations, no evaluable vibrations could be detected using this sensor.

### 3.2. Demographics. Comparison of Watchkeepers and Day Workers

The total sample consisted of 19 male seafarers with a mean age of 42.2 years who had been on board for an average of 115.1 days at the time of the PSG measurement (Table 4). It consisted mostly of non-Europeans (73.7%) and was composed of 13 ratings and 6 officers. Furthermore, the study collective was divided into eleven watchkeepers and eight day workers. Both groups showed a tendency toward pre-obesity (mean BMI ≥ 25 kg/m^2^) and smoking (currently or previously) in about 50%. In addition, about half of the seafarers felt disturbed by noise and vibration in everyday shipboard life, and one-third felt disturbed by the ship’s movements.

The subjective sleep parameters of the total collective showed that 86.7% of the participants described themselves as snorers (Table 5). Moreover, one-third of the seafarers stated that they suffered from sleep problems.

The subjective sleep parameters, as well as subjective and objective sleepiness, showed no significant differences when comparing watchkeepers and day workers. The subjective sleepiness measurement by ESS indicated a tendency toward a higher value of the day workers with a median of 9 (range 3–12) compared to a median ESS value of 6 (range 3–11) of the watchkeepers. Both groups exceeded the threshold of increased normal daytime sleepiness of >5 on average [29]. The day workers even achieved an ESS > 10 in 28.6% of the cases, which indicates the presence of excessive daytime sleepiness. The mean rPUI indicates normal values only in 55.6%. Two watchkeepers (33.3%) and one day worker (33.3%) were unfit for duty according to the rPUI threshold.

However, the evaluation of the PSG parameters for watchkeepers and day workers revealed significant differences (Table 6). Watchkeepers showed a significantly shorter TIB and TST, as well as an increased sleep stage transition index and wake number index. Furthermore, a tendency toward a lower sleep efficiency (SE%), longer deep sleep latency, and more WASO (wakefulness after sleep onset), as well as a higher arousal index, could be observed in the group of the watchkeepers.

Watchkeepers and day workers did not differ significantly with regard to their OSA diagnosis. In general, obstructive apneas were predominant. Mixed and central apneas were detected to a lesser extent. The dominant sleeping position was the supine position, with more than 60%, during which a higher RDI (respiratory disturbance index) could be detected in both groups.

### 3.3. Prevalence of Sleep Disordered-Breathing in Seafarers

Of the 19 seafarers studied, 73.7% showed an AHI ≥ 5, 47.4% had an AHI ≥ 15, and 15.8% an AHI ≥ 30. An AHI ≥ 5 associated with an ESS score > 10 occurred in 16.7% of cases. This means that only 21.4% of the seafarers with an objectively measured AHI ≥ 5 also subjectively suffered from excessive daytime sleepiness (EDS) (Table 7). 

### 3.4. Comparison of the Objective Sleep Quality of Seafarers with the Normal Population

To evaluate the polysomnography parameters of the seafarers measured on board, results of the normal population of a meta-analysis by Boulos et al. [30] were used as a reference (Table 8). This comparison showed that the studied seafarers had a significantly shorter TST (*p* = 0.022). The distribution of sleep stages also demonstrated that the N1 stage occurred significantly more often (*p* < 0.001), and the REM stage occurred significantly less often (*p* < 0.001) in seafarers than in the normal male population (Figure 2).

In addition, a significantly higher arousal index was observed in the seafarer collective (*p* = 0.003). However, a shorter wake time after sleep onset (WASO; *p* = 0.013) was measured in the seafarers. Furthermore, a significantly higher apnea–hypopnea index (AHI; *p* = 0.011) and lower minimum oxygen saturation (minimum SpO_2_; *p* = 0.008) were observed among seafarers. Other parameters, such as sleep efficiency (SE%), percentage N2 and N3 of total sleep time, latency to fall asleep, REM latency, as well as mean SpO_2_ and PLMS index (periodic leg movements in sleep), showed no significant differences, taking into account the small sample size.

### 3.5. The Influence of Age, Seafaring Experience and Length of Stay on Board on Sleepiness and Sleep Behavior 

In order to assess the effects of age and seafaring experience of the seafarers, as well as the duration of stay on board at the time of the polysomnographic measurement, an analysis of the correlation of these parameters with polysomnographic parameters (parameters according to Table 6) as well as the ESS and rPUI values was carried out.

Similar significant correlations to the PSG parameters were found for the seafaring experience and the age of the seafarers (Table 9a). Firstly, the experienced and older seafarers showed longer TIB and TST. Secondly, the AHI, RDI, obstructive apnea index, and the number of respiratory-related waking events per hour (#wake respiratory index) correlated positively with the age and experience of the seafarers.

The length of stay on board, on the other hand, showed a strong positive significant correlation with the percentage of sleep in the supine position as well as a significant negative correlation with the TIB and a marginal significance with the TST (Table 9a).

No significant correlation was observed between subjective or objective sleepiness (ESS and rPUI) with seafaring experience, age, or length of stay on board (Table 9b).

### 3.6. Correlation of Sleepiness on Board Measured with ESS, Pupillometry and PSG

In order to assess whether the sleepiness measured by ESS and pupillometry was also reflected in the PSG recording, sleep onset latency (SOL) was used as a comparative parameter.

The ESS value as a measure of subjective sleepiness and the rPUI value as a measure of the objective sleepiness effect showed a significant, strong positive correlation with each other (r = 0.667; *p* = 0.049). In contrast, only a non-significant, medium–strong positive correlation was found between SOL and ESS values (r = 0.415; *p* = 0.087). The correlation between SOL and rPUI values was also not significant and weak (r = 0.117; *p* = 0.765).

In addition, it was tested whether an increased arousal index as a possible cause of sleepiness correlated with the ESS and rPUI values. However, no significant correlation of the ESS or rPUI with the arousal index was found.

## 4. Discussion

The results of this pilot study indicate that this research is a valuable addition regarding the knowledge of the technical assessment of the sleep architecture of seafarers on the high seas, also considering the duration of embarkment and the age of the seafarers. In addition, the findings may contribute to cardiovascular prevention in the pre-stressed group of seafarers.

### 4.1. Quality of Polysomnography on Board Merchant Ships

The good signal quality of the polysomnographies measured on board as well as the good to sufficient impedances speak for a high quality of the polysomnographic measurements performed on the high seas, which in our experience were comparable with measurements in a sleep laboratory on land. The lack of global artifacts and sleep stage artifacts also showed that the onboard environment did not fundamentally disturb the polysomnography (PSG). The latter is dependent on EEG and EMG electrodes and confirms the reliable feasibility of these measurements on board.

In studies, an artifact-free time of 75% of the recording time of individual leads is often given as a criterion for a well-interpretable PSG [32,33]. The percentage of oxygen (O_2_), as well as heart rate (HR) artifacts, did not exceed the critical value of 25% artifact percentage during the present recording. The proportion of respiratory artifacts exceeded 25% on average, but this value was due to a single outlier because of the small study sample. Thus, no relevant disturbances of the individual polysomnography components can be reported. Accordingly, it can be assumed that polysomnography on board and on the high seas is of good feasibility and quality.

### 4.2. Comparison of Watchkeepers and Day Workers

The tendency for day workers to have higher scores on the Epworth Sleepiness Scale (ESS) on average contradicts our initial hypothesis as well as previous maritime studies that found significantly more often daytime sleepiness among watchkeepers [5]. However, Oldenburg and Jensen [4] did not observe a significant difference in ESS scores between these groups. In addition, the small sample size may have contributed to this result in the present study.

The partly significant difference between watchkeepers having poorer sleep quality with shorter total sleep time (TST), more sleep stage transitions, longer deep sleep latency, and a greater proportion of awake time or sleep fragmentation due to arousals suggests that the objective sleep quality of watchkeepers on board is poorer compared to day workers. This was also reflected in the lower sleep efficiency (SE%) of watchkeepers, which at 78.3% (SD 11.5) was below the common threshold of 85–90% [26], while day workers had a SE% of 87.0% (SD 5.7). In contrast, Oldenburg and Jensen [4] reported a generally lower SE% of watchkeepers (67.9% (SD 12.2%)) and day workers (72.7% (SD 11.8%)) in actigraphy measurements. Even though actigraphs are supposed to represent TST, sleep onset latency (SOL) and SE% measured by PSG with good agreement, overestimation of sleep period and underestimation of wakefulness were observed more frequently with actigraphic measurements [9]. Based on these findings, one would have expected an even lower polysomnographically measured SE% in our collective. However, due to the small collective and the one-time measurement, random bias of the results cannot be excluded. Further studies with a larger collective and a higher number of PSG measurements on board are, therefore, recommended.

Since the apnea–hypopnea index (AHI), the obstructive sleep apnea (OSA) diagnosis, as well as the PLMS index (periodic leg movements in sleep), hardly differed between watchkeepers and day workers, it is likely that the different working conditions of watchkeepers and day workers contributed decisively to the partly significantly changed sleep architecture. The guideline “Health Aspects and Design of Night and Shift Work” of the German Society for Occupational and Environmental Medicine [34] states that irregular shift times are increasingly associated with sleep deficits and fractionated sleep periods.

It was striking that watchkeepers, as well as day workers, slept significantly more often in the supine position (>60%) compared to a study of the normal population, in which men slept 55.6% (SD 17.8%) on their sides and only 35.1% (SD 18.2%) on their backs [35]. It should be mentioned here that in this present maritime study, the percentage of TST was assessed, whereas, in the study of the normal population, the percentage of time in bed (TIB) was evaluated. In addition, it was observed that poor sleepers spend more time in the supine position [36]. These statements could indicate that seafarers prefer the supine position significantly more often due to poor sleep quality. Further studies on this topic would be interesting, especially since sleeping in a supine position may promote OSA [10]. The increased supine position in combination with an increased respiratory disturbance index (RDI) also suggests that avoiding the supine position (e.g., through a sleeping backpack) as a therapeutic approach for OSA could also be considered in the seafarer population as a cost-effective and easily implemented prevention option.

### 4.3. Prevalence of Sleep-Disordered Breathing in Seafarers

Comparing the prevalence of obstructive sleep apnea (OSA) based on different apnea–hypopnea index (AHI) values of our seafarer collective with that of the normal population turned out to be difficult, as sometimes large discrepancies between individual prevalence studies could be found.

In a review that included eleven epidemiological studies from 1993 to 2013 from different countries regarding AHI, an average male OSA prevalence of 22% was observed, varying widely (9–37%) and increasing in more recent studies. Different diagnostic devices, AHI scoring criteria, study designs, and characteristics of the included subjects have been discussed as possible reasons for these different results [37].

A large-scale Swiss study from 2009–2013 indicated that 83.3% of male subjects from the normal population showed an AHI ≥ 5, and 49.7% had an AHI ≥ 15 [23]. Following this reference, the OSA prevalence of seafarers would actually be lower when measured by the elevated AHI (AHI ≥ 5 = 73.7%; AHI ≥ 15 = 47.4%). However, this discrepancy could also be due to the higher median age in the Swiss study population of 56 years (40–60 years) compared to the median age of the seafarers of 42.2 years (25–61 years), as OSA prevalence or AHI increases with age [37]. Therefore, the actual AHI values or OSA prevalence of the seafarer collective could, in reality, be the same or even higher compared to the normal population, especially considering the healthy worker effect (due to the seafarers’ regular medical fitness test for nautical service every two years), which basically assumes an underestimation of work-related morbidity [18]. In addition, the significantly increased AHI of seafarers compared to the Boulos et al. [30] collective, discussed below, would suggest an increased OSA prevalence among seafarers. However, it should be kept in mind that in this current study, mainly the snorers were motivated to participate. This may have contributed to an increase in detected OSA prevalence due to selection.

The prevalence of co-occurrence of AHI ≥ 5 and ESS > 10, which is used in many studies as a diagnostic criterion for obstructive sleep apnea syndrome (OSAS), was slightly increased in the seafarers with 16.7% (n = 3) compared to the normal male population with 12.5% [38]. This could be an indication that the working and living conditions on board increase the symptoms of excessive daytime sleepiness (EDS) in OSA patients. Even though only 21.4% of seafarers affected by OSA were found to have EDS and OSA and EDS were not strictly correlated [10], the impact of OSA resulting in EDS should not be underestimated. For example, OSA patients with increased daytime sleepiness were found to be three to seven times more likely to have road traffic accidents [10]. In addition, cardiovascular diseases, in particular strokes, are associated with OSA. Furthermore, an increased risk of early death was observed in OSA patients under 70 years of age [37]. Thus, intervention for seafarers is advisable not only from the aspect of onboard safety but also from a health perspective. In particular, the increased risk of cardiovascular disease in untreated OSA is an argument for screening seafarers for this disease since an increased tendency to cardiovascular disease in seafarers is discussed already [18]. Moreover, the seafarer population fulfills a large part of the OSA risk factors [10]: male gender, increased body mass index (BMI) [18], as well as increased alcohol and cigarette consumption [39].

In this context, it should also be considered that, for example, an increased BMI (which in our collective was >30 on average) is also associated with short sleep periods, as studies showed a consistent relationship between short sleep duration and higher total energy as well as fat intake [40]. In a study by Neumann et al. [41], 43.9% of seafarers stated that they had gained weight since practicing this profession. Therefore, the codependence of lifestyle and sleep parameters should also be investigated in future studies.

Since some stress factors, which are presumably related to disturbed sleep as well as cardiovascular diseases (e.g., shift work, noise, and vibrations), can only be reduced to a limited extent [18], OSA as an additional risk factor for seafarers should be diagnosed and treated if possible.

Loud snoring is not considered a sure sign of OSA [42] but could be an important screening parameter to specifically screen seafarers for sleep-disordered breathing. This recommendation was also made by Mäkelä and Savolainen [43], who demonstrated a diagnosis of OSA in one-fifth of loudly snoring military service members. Another common way to pre-select OSA patients is the STOP-Bang questionnaire, which showed a sensitivity and specificity of more than 90% in patients with moderate to severe OSA [44].

### 4.4. Comparison of the Objective Sleep Quality of Seafarers with the Normal Population

The sleep parameters from the study of Boulos et al. [30]—which represented the normal population in our comparison—were based on healthy males (18 years and older). This group could be compared to our collective so far as the seafarers also consisted exclusively of male adult participants who underwent a health check every two years. It might be possible that the sleep quality and sleep-related respiratory events in seafarers cannot be entirely compared to the general population of Boulos et al. [30] since they excluded subjects with a BMI > 30 or sleep problems as well as shift workers.

The comparison of the polysomnographic sleep parameters of seafarers with reference parameters of the normal population showed that seafarers had a significantly lower average total sleep time (TST). At 5.4 h (323.2 min), it was also below the minimum sleep duration of 6 h recommended by the National Sleep Foundation [45]. However, it should be noted that the assessment of total sleep time is not necessarily decisive for the evaluation of sufficient, restful sleep. For this purpose, the subjective perception of the person in the waking state should also be taken into account [46].

Another parameter for the assessment of sleep quality is the arousal index. Although there is no clear cut-off value for a pathologically increased arousal index, there are proposals for a definition of an increased arousal index at >10 arousals per hour [47]. According to this definition, both the seafarer collective and the normal population would have an increased arousal index, but a comparison of these groups showed a significantly higher arousal index among the seafarers, which at 29.4 (95% CI of 20.4–38.4) exceeded the threshold value of 10 arousals per hour on average almost threefold. Arousals cause fragmentation of sleep, which can have an effect similar to sleep deprivation. It has been shown that EEG arousals could lead to an increase in subjective and objective daytime sleepiness as well as a reduction in psychomotor performance [48]. A detailed arousal analysis also revealed that the arousal index consisted of a large extent of respiratory arousals (mean 10.5/h; SD 14.6/h). This fact, in addition to the increased AHI, which will be discussed in more detail below, is an indication of the increased occurrence of pathological respiratory events among seafarers. Whether the increased arousal index was also generally caused by disturbing external stimuli (e.g., ambient light, noise, vibration, ship movement) cannot be differentiated without additional environmental measurements.

Furthermore, more superficial N1 sleep occurred on board, which can be seen as a transition between wakefulness and sleep and is not very restful. An increased N1 proportion is also indicative of sleep fragmentation due to arousals, which can be attributed primarily to sleep disorders, such as sleep apnea and periodic sleep movements, snoring, and disturbing environmental influences [49].

In the seafarer collective studied, the mean PLMS index (periodic leg movements in sleep), also similar to the PLMS index of the reference study, was not only below the current clinically relevant value of 15 PLMS per hour but also below the former cut-off value of five PLMS per hour [50]. In the arousal analysis, only a very small proportion of PLMS-induced arousals (mean 0.6/h; SD 1.2/h) could be detected. The exact reasons for the increased occurrence of the arousals as well as sleep stage N1 in seafarers should be critically questioned with regard to the health of the crew and safety on board and investigated more closely in subsequent studies.

In addition, a lower percentage of REM sleep (rapid eye movement) in the total sleep of seafarers was found. The actual function of REM sleep is not yet fully understood, and the shortening of the REM phase also offers potential for discussion. Primarily, REM sleep is thought to have an important function in processes of psychological well-being, intellectual performance, as well as procedural memory [46]. Too little REM could, therefore, have a negative impact on the mental state of seafarers, who are already exposed to extreme stress [1]. Secondly, an increased risk of accidents on board would be conceivable due to the reduced ability to comprehend. However, studies on REM deprivation in the normal population have so far shown controversial results. Both increased daytime sleepiness by means of a multiple sleep latency test (MSLT) with a REM reduction to 9% of the TST [51], as well as unchanged daytime sleepiness (unchanged MSLT) with a reduction of the REM portion to 4.5% [52], could be proven. As a possible explanation for the unchanged daytime sleepiness during strong REM deprivation, an increased central nervous system excitability was discussed as a compensatory mechanism [52]. Comparing these observations with the mean REM proportion of the seafarer collective of 13.7% (95% CI of 10.9–16.4%), there could be a risk of increased daytime sleepiness due to the reduced REM proportion.

The significant differences in the N1 proportion and arousal index already discussed contrast with the shorter wake-after-sleep onset (WASO) time of the seafarers compared to the reference group. It should be kept in mind that the WASO parameter is not a relative value, but it naturally becomes lower when the total sleep time (TST) is shortened, as was the case in our collective. From this point of view, the WASO expressed in minutes does not seem to be fully recommendable when comparing collectives with significantly different total sleep times.

Furthermore, there was no significant difference in sleep efficiency (SE%) between seafarers and the reference study. According to the literature, a SE% of 85–90% is considered good [26]. Thus, both the seafarers’ SE% of 81.9% (95% CI of 77.0–86.9) and the normal population’s SE% of 84.3% (95% CI of 82.0–86.6) could be considered as lowered. However, it should be kept in mind that the SE% only indicates the percentage of the time in bed (TIB) in which the person is actually in a sleep state. Thus, not all arousals are necessarily included in the calculation of the SE%, as these do not always lead to a complete waking reaction [26]. Accordingly, the significantly increased arousal index of seafarers suggests that sleep quality assessment should not rely solely on sleep efficiency, which is used as a common sleep quality measurement parameter in actigraphic measurements on board ships [3,4,53,54,55].

The significantly elevated apnea–hypopnea index (AHI) could be indicative of an increased risk of sleep-related breathing disorders in seafarers. The average AHI of 18.2 (95% CI of 8.6–27.9) in the seafarer collective, in combination with clinical symptoms and certain comorbidities, could already be evidence of moderate-grade obstructive sleep apnea (OSA) [10]. In studies of healthy volunteers on land, both a significant increase in the AHI [56] and increased collapsibility of the upper airways [57] were found during nights with sleep fragmentation. In addition, the significantly lowered minimum oxygen saturation also speaks for respiratory restrictions in the seafarers examined. Although there was no clear cut-off value for reduced oxygen saturation, the mean SpO_2_ value of 94.7% (95% CI of 93.6–95.9) was at the 95% limit, which was used as a cut-off value in many studies [58]. The minimum SpO_2_ was significantly lower at 82.5% (95% CI of 78.7–86.3). In this regard, it should be noted that 52.6% of the subjects who were smokers or former smokers were at increased risk for a reduced SpO_2_ value [59]. The actual connection of sleep-related breathing events with fragmented sleep of seafarers as well as the working and sleeping environment on board, should be investigated more deeply on this basis.

### 4.5. Influence of Age, Seafaring Experience, and Length of Stay on Board on Sleepiness and Sleep Behavior

The specific investigation of the age and seafaring experience of the seafarers in connection with polysomnographic parameters showed that age and experience are closely linked, as was expected. The increase in the respiratory parameters apnea–hypopnea index (AHI), respiratory disturbance index (RDI), obstructive apnea index, as well as the number of respiratory-related waking events per hour (#wake respiratory index) can be explained by a general rise in the prevalence of such events with increasing age [37]. However, according to the reference study, the total sleep time (TST) should decrease with age [30]. The fact that the time in bed (TIB) and TST in our study increase with the age and experience of the seafarers could, on the one hand, speak for an increased need for recovery of the seafarers in higher age and, on the other hand, for the fact that they could integrate longer recovery periods into their daily routine due to more work experience.

With a longer stay on board, the seafarer collective showed a significantly shorter TIB as well as a tendency toward a lower TST. In contrast, Hystad and Eid [60] observed only a limited effect of duration at sea on fatigue and subjective sleep quality. However, it should be borne in mind that the present study design only depicts a snapshot of the seafarers’ sleep, and thus, no changes over time can be assessed. From this point of view, the evaluation of the influence of the length of stay on board on the sleeping position as well as the increased supine position is only of limited significance, as we do not know whether a seafarer already slept more on his back at the beginning of his stay on board. These results and their possible causes should, therefore, be examined in more depth in longitudinal studies.

### 4.6. Correlation of Sleepiness on Board Measured with ESS, Pupillometry and PSG

Even though the multiple sleep latency test (MSLT) is no longer considered the gold standard of sleepiness diagnostics, this test, which uses a shortened sleep onset latency (SOL) as a measure of sleepiness, is still frequently used [26]. However, this method does not seem to be very practicable for everyday work on board due to the test execution with five short daytime sleep units in a two-hour interval following a polysomnographically recorded night. We, therefore, investigated whether, with the help of the SOL from a PSG measurement, a statement can be made about the daytime sleepiness of the sailors, which was measured subjectively by means of the ESS and objectively by means of pupillometry.

In our study, only a significant correlation between the ESS and pupillometry (relative pupillary unrest index; rPUI) was found. A shortened SOL, on the other hand, only showed coinciding tendencies toward increased ESS and rPUI values but no significance. Thus, no recommendation can be made for the use of SOL measured by PSG as a sole sleepiness indicator.

However, the fact that pupillometry was not measured contemporaneously to PSG in all cases limits the validity of this comparison. In addition, the one-time objective measurements only provide a momentary actual state. Therefore, these are not necessarily representative ways of comparing the assessments in the ESS questionnaires, which refer to a longer period of several weeks.

Moreover, it is discussed whether and to what extent subjective sleepiness correlates with objective measurements. Oldenburg and Jensen [4] found only a weak correlation between the subjective Stanford Sleepiness Scale (SSS) scores and seafarers’ pupillometry. In the study by Yamamoto et al. [61], no correlation was found between the ESS and objective sleepiness measurement methods (SOL of the MSLT, rPUI, and PUI). However, the rPUI showed a correlation with the SOL of the MSLT. Aurora et al. [62], in contrast, found a clear correlation between ESS and SOL of the MSLT. Nevertheless, the authors recommended increasing the most frequently used cut-off value of the ESS questionnaire from >10 to ≥13 to achieve a better match with objective sleepiness.

These discrepancies show that the different methods of measuring sleepiness must be critically questioned and cannot be compared arbitrarily. It should be noted that different aspects of sleepiness are represented. The ESS measures global subjective sleepiness, while objective measures (MSLT and pupillometry) reflect the sleepiness component of tonic activation (the general level of alertness) [26].

In summary, sleepiness measurement offers great potential for discussion. According to our research, the SOL of a PSG measurement is not suitable for making a valid statement about the daytime sleepiness of seafarers on board. A combination of a questionnaire (e.g., ESS) and pupillometry still seems to be preferable for this purpose.

Furthermore, contrary to expectations, the arousal index correlated negatively with the ESS and rPUI scores. Even though this result was not significant, it indicates that an increased arousal index does not necessarily lead to subjective or objective sleepiness.

## 5. Limitations

In addition to the possible impacts on the results due to the pilot study design with a small study population, a selection bias is possible, as snorers were specifically approached for this study. Moreover, it cannot be ruled out that seafarers with sleep problems were more likely to participate in this study. In addition, the impact of a first-night effect (FNE) of the polysomnographic measurements should be considered, which can lead to prolonged SOL and REM latency, increased light sleep (N1) and waking episodes, as well as reduced deep sleep and REM sleep [10]. However, the actual impact of an FNE in ambulatory PSG is controversial and probably not as pronounced as in polysomnographic measurements in a sleep laboratory [9].

## 6. Conclusions

The polysomnographies (PSGs) conducted on board merchant ships on the high seas in this study showed signal qualities and impedances comparable to a sleep laboratory. Conspicuous artifacts, which could be attributed to the ship environment, were not detected. Therefore, good feasibility and quality of the polysomnography on board can be assumed.

The comparison of the sleep architecture of seafarers with the normal population provided evidence that not only the total sleep time on board was lower but also that changes in sleep occurred in the macroarchitecture (shift of deep sleep phases in favor of light sleep phases) as well as the microarchitecture (increased arousal index). However, no statement can be made about reasons why individual sleep parameters differed significantly from the normal population, as no parallel environmental measurements were taken on board, and the one-time polysomnographic measurement only represented a snapshot. Therefore, further PSG studies are recommended to examine the sleep architecture of seafarers or even passengers on board over several nights in comparison to land-based measurements. In addition, such studies should be conducted in connection with environmental measurements (e.g., room climate, poor bedding conditions, ambient light, vibration, noise, or ship motion) as well as lifestyle parameters (e.g., BMI, nutritional behavior, and physical activity).

Seafarers have an increased risk profile for Obstructive Sleep Apnea (OSA). In fact, 73.7% of seafarers were diagnosed with at least mild OSA (AHI ≥ 5), 15.8% with severe OSA (AHI ≥ 30), and 16.7% with additional excessive daytime sleepiness (AHI ≥ 5 + ESS > 10). A mildly elevated prevalence of OSA among seafarers is possible, even though the comparison with the normal population was found to be difficult. Nevertheless, additional studies investigating OSA in seafarers and its effects on daytime sleepiness and general health, especially with regard to cardiovascular disease, seem useful.

Furthermore, poorer objective sleep quality was found among watchkeepers. In general, seafarers slept conspicuously, often in the supine position. This sleeping position also correlated with a longer stay on board. The possible causes and effects of this observation should be investigated more closely in future long-term studies with a larger collective.

In general, it must be emphasized that the results of this pilot study, although it included only a small number of subjects, on the one hand, provided valuable insights into the measurability of seafarers’ sleep architecture for future studies. On the other hand, the data indicated the importance of seafarers’ poor sleep quality and OSA risks. Therefore, the need for larger long-term studies is highly suggested.

## Figures and Tables

**Figure 1 ijerph-20-03168-f001:**
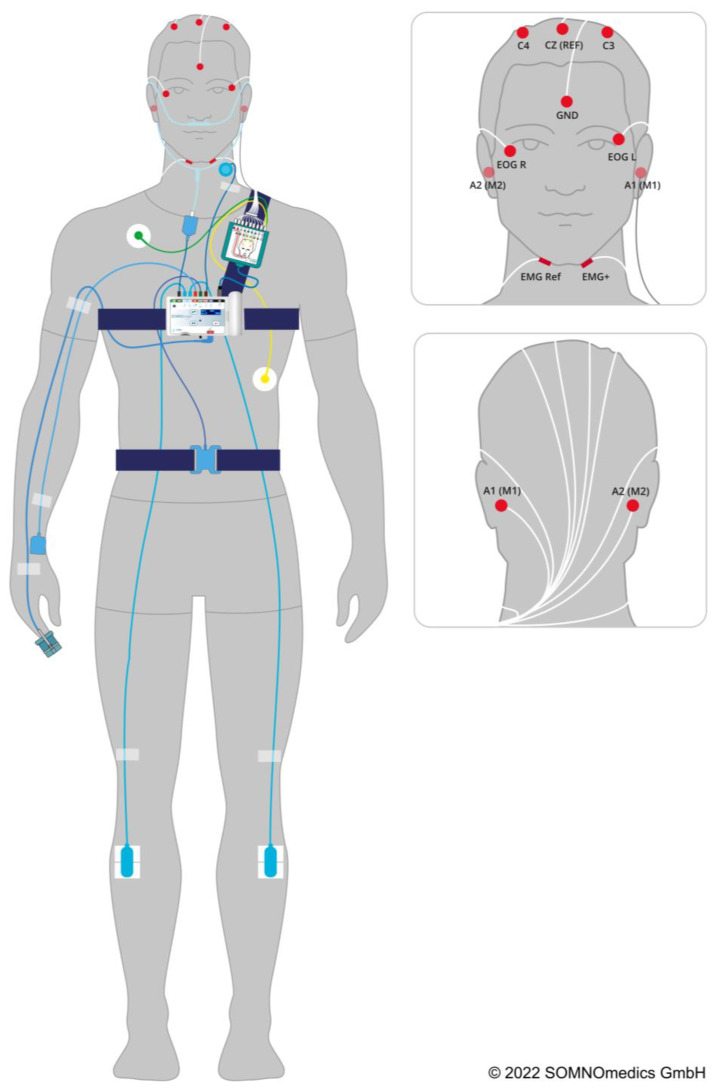
Schematic overview of the polysomnographic leads. © 2023 SOMNOmedics GmbH.

**Figure 2 ijerph-20-03168-f002:**
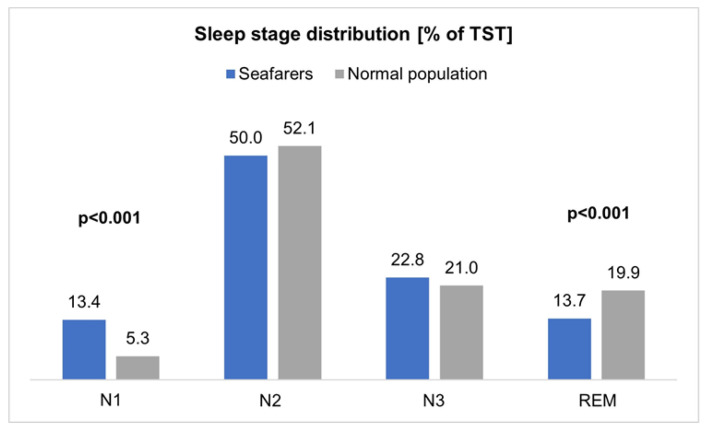
Percentage sleep stage distribution of seafarers and the reference values of the normal male population (age 18 years and older) according to Boulos et al. [30].

**Table 1 ijerph-20-03168-t001:** Overview of TIB during the PSG recordings divided into watchkeepers and day workers.

	Start of TIB	Number	Percentage
Watchkeepers (n = 11)	midnight–4 a.m.	3	27.3
	4 a.m.–8 a.m.	2	18.2
	8 p.m.–midnight	6	54.5
Day workers (n = 8)	8 p.m.–midnight	8	100.0

PSG = polysomnography; TIB = time in bed.

**Table 2 ijerph-20-03168-t002:** Impedance and signal quality of the polysomnograms.

	Impedance Quality n (%)	Signal Quality n (%)
Good	11 (57.9)	19 (100.0)
Sufficient	8 (42.1)	0 (0)
Poor	0 (0)	0 (0)

**Table 3 ijerph-20-03168-t003:** Artifacts of the polysomnographic measurements on board.

	n (%)	Mean	SD	Minimum	Maximum
Resp. artifacts [%TIB]	2 (10.5)	30.5	40.1	2.1	58.8
O_2_-artifacts [%TIB]	7 (36.8)	6.7	6.3	0.1	16.9
HR-artifacts [%TIB]	3 (15.8)	0.2	0.1	0.1	0.2

resp. = respiratory; O_2_ = oxygen; HR = heart rate; TIB = time in bed.

**Table 4 ijerph-20-03168-t004:** Demographic and lifestyle parameters in a comparison of watchkeepers and day workers.

	Total (n = 19)	Watchkeepers (n = 11)	Day Workers (n = 8)
Demographic and lifestyle parameters			
Age, mean (min–max)	42.2 (25–61)	41.6 (25–61)	43.1 (27–60)
Nationality, n (%)			
European	5 (26.3)	3 (27.3)	2 (25.0)
Non-European	14 (73.7)	8 (72.7)	6 (75.0)
Officers, n (%)	6 (31.6)	4 (36.4)	2 (25.0)
Non-Officers, n (%)	13 (68.4)	7 (63.6)	6 (75.0)
Length of stay on board [d], mean (SD)	115.1 (65.0)	106.1 (65.1)	127.4 (67.3)
BMI [kg/m^2^], mean (SD)	27.2 (4.4)	27.3 (4.1)	27.2 (5.0)
BMI ≥ 25, n (%)	11 (57.9)	7 (63.6)	4 (50.0)
Smoking (current or former), n (%)	10 (52.6)	6 (54.5)	4 (50.0)
Subjectively disturbed on board by:			
Noise, n (%)	10 (52.6)	5 (45.5)	5 (62.5)
Vibration, n (%)	10 (52.6)	5 (45.5)	5 (62.5)
Ship movement, n (%)	7 (36.8)	4 (36.4)	3 (37.5)

A comparison of watchkeepers and day workers in terms of their demographic and lifestyle parameters revealed no significant differences.

**Table 5 ijerph-20-03168-t005:** Subjective sleep parameters and sleepiness of watchkeepers and day workers.

Subjective Sleep Parameters	Total (n = 19)	Watchkeepers (n = 11)	Day Workers (n = 8)	*p*-Value
Snoring, n (%)	13 (86.7)	8 (88.9)	5 (83.3)	1.000 ^3^
Sleep problems, n (%)	6 (31.6)	4 (36.4)	2 (25.0)	1.000 ^3^
Sleepiness				
ESS, median (min–max)	6 (3–12)	6 (3–11)	9 (3–12)	0.536 ^2^
ESS > 5, n (%)	11 (61.1)	7 (63.6)	4 (57.1)	1.000 ^3^
ESS > 10 (EDS), n (%)	3 (16.7)	1 (9.1)	2 (28.6)	0.528 ^3^
rPUI, mean (SD)	1.2 (0.7)	1.1 (0.8)	1.4 (0.7)	0.690 ^1^
rPUI, n (%)				
Normal (<1.02)	5 (55.6)	3 (50.0)	2 (66.7)	
Conspicuous (≥1.02–<1.53)	1 (11.1)	1 (16.7)	0 (0.0)	
Unift for duty (≥1.53)	3 (33.3)	2 (33.3)	1 (33.3)	

^1^ *t*-test for independent samples; ^2^ Mann–Whitney U-Test; ^3^ Fisher’s exact test. Percentages may be based on different populations due to missing values. ESS = Epworth Sleepiness Scale; rPUI = relative pupillary unrest index.

**Table 6 ijerph-20-03168-t006:** Polysomnographic sleep parameters of watchkeepers and day workers.

	Total (n = 19)	Watchkeepers (n = 11)	Day Workers (n = 8)	*p*-Value
Objective sleep quality				
TIB [min], mean (SD)	391.0 (81.5)	359.3 (83.1)	434.7 (58.8)	**0.043** ^1^
TST [min], mean (SD)	323.2 (89.5)	283.3 (90.3)	377.9 (55.1)	**0.018** ^1^
WASO [min], mean (SD)	38.2 (21.6)	41.9 (25.2)	33.1 (15.6)	0.394 ^1^
Sleep efficiency [%], mean (SD)	81.9 (10.3)	78.3 (11.5)	87.0 (5.7)	0.065 ^1^
Sleep onset latency (SOL) [min], mean (SD)	20.9 (26.8)	21.7 (29.9)	19.9 (23.9)	0.840 ^2^
Deep sleep latency (N3 latency) [min], mean (SD)	42.3 (36.5)	46.8 (40.4)	36.3 (32.0)	0.442 ^2^
REM latency [min]	112.9 (59.4)	113.6 (74.2)	111.9 (34.6)	0.946 ^1^
Sleep stage duration [%TST], mean (SD)				
N1	13.4 (7.3)	14.6 (6.0)	11.9 (8.9)	0.446 ^1^
N2	50.0 (11.9)	48.2 (12.3)	52.6 (11.5)	0.437 ^1^
N3	22.8 (11.2)	23.5 (9.0)	22.0 (14.3)	0.787 ^1^
REM	13.7 (5.7)	13.8 (6.7)	13.5 (4.5)	0.930 ^1^
Sleep stage transition index [n/h], mean (SD)	21.6 (11.1)	25.0 (12.3)	17.1 (4.5)	**0.041** ^2^
Arousal index [n/h], mean (SD)	29.4 (18.7)	32.6 (18.4)	24.9 (19.3)	0.388 ^1^
Wake number index [n/h], mean (SD)	7.9 (7.2)	10.2 (8.8)	5.0 (2.1)	**0.016** ^2^
#wake respiratory index [n/h], mean (SD)	3.4 (7.4)	4.9 (9.5)	1.4 (2.3)	0.351 ^2^
PLMS index [n/h], mean (SD)	4.0 (7.2)	3.1 (3.2)	5.3 (10.8)	0.600 ^2^
Body position change index [n/h], mean (SD)	2.8 (1.8)	3.0 (2.1)	2.5 (1.5)	0.840 ^2^
Respiratory analysis				
AHI [n/h], mean (SD)	18.2 (20.0)	17.8 (20.0)	18.9 (21.2)	0.904 ^2^
AHI [n/h], n (%)				
<5	5 (26.3)	3 (27.3)	2 (25.0)	
5–14	5 (26.3)	3 (27.3)	2 (25.0)	
15–29	6 (31.6)	3 (27.3)	3 (37.5)	
≥30	3 (15.8)	2 (18.2)	1 (12.5)	
RDI [n/h], mean (SD)	20.7 (20.3)	21.8 (20.7)	19.3 (21.1)	0.778 ^2^
Obstructive apnea index [n/h], mean (SD)	11.6 (19.3)	11.2 (19.9)	12.2 (19.9)	0.840 ^2^
Mixed apnea index [n/h], mean (SD)	0.8 (2.6)	1.3 (3.4)	0.09 (0.2)	0.310 ^2^
Central apnea index [n/h], mean (SD)	0.5 (1.5)	0.8 (2.0)	0.06 (0.1)	0.442 ^2^
Snoring duration [min], mean (SD)	27.7 (34.2)	19.34 (21.0)	39.3 (46.0)	0.657 ^2^
Oxygen saturation				
Desaturation index [n/h], mean (SD)	12.0 (18.7)	11.4 (19.7)	12.8 (18.4)	0.657 ^2^
Minimum SpO_2_ [%], mean (SD)	82.5 (7.9)	83.1 (8.5)	81.6 (7.5)	0.702 ^1^
Mean SpO_2_ [%], mean (SD)	94.7 (2.4)	94.6 (2.7)	95.0 (2.1)	0.717 ^2^
Sleep position				
Sleep position [%TST], mean (SD)				
Supine position	63.5 (30.8)	62.7 (35.1)	64.6 (26.0)	0.968 ^2^
Non-supine position	36.5 (30.8)	37.4 (35.1)	35.4 (26.0)	0.968 ^2^
RDI [%TST], mean (SD)				
in supine position	29.4 (26.7)	30.1 (26.5)	28.6 (28.7)	0.904 ^2^
in non-supine position	5.7 (8.2)	5.6 (9.9)	5.8 (5.6)	0.310 ^2^

^1^ *t*-test for independent samples; ^2^ Mann–Whitney U-test; significant findings in bold. TIB = time in bed; TST = total sleep time; WASO = wake after sleep onset; REM = rapid eye movement; PLMS = periodic leg movements in sleep; AHI = apnea–hypopnea index; RDI = respiratory disturbance index.

**Table 7 ijerph-20-03168-t007:** AHI of seafarers (n = 19) by severity level.

AHI Severity, n (%)	AHI ≥ 5	AHI ≥ 15	AHI ≥ 30	AHI ≥ 5 + ESS > 10 ^1^
	14 (73.7)	9 (47.4)	3 (15.8)	3 (16.7)

AHI = apnea–hypopnea index; ESS = Epworth Sleepiness Scale. ^1^ Results based on n = 18 seafarers.

**Table 8 ijerph-20-03168-t008:** Comparison of polysomnography parameters of seafarers with reference values from the normal male population (age 18 years and older).

	Seafarers (Males, n = 19)	Reference Parameters ^1^ (Males)	*p*-Value ^2^
	Mean (95% CI)	Mean (95% CI)	
TST [min]	323.2 (280.0–366.3)	374.6 (357.3–392.0)	**0.022**
SE [%]	81.9 (77.0–86.9)	84.3 (82.0–86.6)	0.329
WASO [min]	38.2 (27.8–48.6)	51.8 (42.1–61.4)	**0.013**
Sleep onset latency (SOL) [min]	20.9 (8.0–33.9)	14.7 (13.0–16.4)	0.324
REM latency [min]	112.9 (84.3–141.5)	92.5 (85.8–99.2)	0.152
Arousal index [n/h]	29.4 (20.4–38.4)	14.5 (12.6–16.5)	**0.003**
AHI [n/h]	18.2 (8.6–27.9)	5.2 (4.2–6.1)	**0.011**
Mean SpO_2_ [%]	94.7 (93.6–95.9)	94.7 (94.3–95.1)	0.947
Minimum SpO_2_ [%]	82.5 (78.7–86.3)	87.9 (86.6–89.2)	**0.008**
PLMS index [n/h]	4.0 (0.5–7.8)	2.1 (1.3–3.0)	0.266

^1^ from Boulos et al. [30]; analyzed according to AASM criteria 2007 [31] or 2012 [24]; significant findings in bold. ^2^ Evaluated using one-sample *t*-test.

**Table 9 ijerph-20-03168-t009:** (**a**) Correlation of age, seafaring experience, and length of stay on board of the seafarers with polysomnographic parameters. (**b**) Correlation of age, seafaring experience, and length of stay on board with subjective and objective sleepiness.

**(a)**							
	**AHI**	**RDI**	**#Wake Resp. Index**	**Obstr. Apnea Index**	**TIB**	**TST**	**Supine Position**
Age	r = 0.537 *p* = **0.018**	r = 0.504 *p* = **0.028**	r = 0.481 *p* = **0.037**	r = 0.474 *p* = **0.040**	r = 0.542 *p* = **0.016**	r = 0.489 *p* = **0.034**	r = 0.002 *p* = 0.994
Seafaring experience (in years)	r = 0.509 *p* = **0.026**	r = 0.515 *p* = **0.024**	r = 0.508 *p* = **0.026**	r = 0.480 *p* = **0.037**	r = 0.537 *p* = **0.018**	r = 0.536 *p* = **0.018**	r = −0.124 *p* = 0.614
Length of stay on board	r = 0.191 *p* = 0.433	r = 0.181 *p* = 0.459	r = 0.128 *p* = 0.600	r = 0.126 *p* = 0.608	r = −0.520 *p* = **0.022**	r = −0.443 *p* = 0.058	r = 0.639 *p* = **0.003**
**(b)**							
				**ESS**		**rPUI**	
Age				r = −0.078 *p* = 0.757		r = −0.333 *p* = 0.381	
Seafaring experience (in years)				r = −0.233 *p* = 0.352		r = −0.238 *p* = 0.537	
Length of stay on board				r = 0.156 *p* = 0.536		r = 0.323 *p* = 0.397	

Significant findings in bold.

## Data Availability

Not applicable.

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
