# Peer review of "Sleep Architecture and Sleep-Related Breathing Disorders of Seafarers on Board Merchant Ships: A Polysomnographic Pilot Field Study on the High Seas"

_ijerph, 2023, doi:10.3390/ijerph20043168_

Round 1
Reviewer 1 Report
Dear Authors,
thanks a lot for your well-written and valuable article. It deals with a relevant topic for seafrares and prevention of well-being, safety and mental-cardiovascular diseases. The main question addressed by the research is the feasibility and quality of sleeping of seafarers on board merchant ships.
This article is well structured, methods are adequate, results so and you perform very good discussion and conclusion
I have only few comments.
you used several acronyms along your paper. I think, as you did for some, it is ueful to rewrote the entire word and his acronym in () first time it is reused in discussion. like l 536-38 be consistent with l 581
Specific Points:
1. It is the first study with polysomnography onboard and it reveals impact of lenght of embarkment, age. It is important for sleeping and cardiovascular diseases prevention
2. It adds new knowledge on sleeping architecture, the feasibility of technical assessment of sleeping on board ships. Seafarers have higher risks of obstructive sleep apnea.
3. Methods are well, nothing more needed.
4. Sometimes the Discussion is not easy to follow with a lot of abbreviations and some parts are very technical.
best regards
Author Response
We would like to thank the reviewer for the helpful comments that are valuable for improving the quality of the present manuscript. In this response, the reviewers’ comments are presented in bold italic letters and followed by our answers:
Reviewer 1
“you used several acronyms along your paper. I think, as you did for some, it is useful to rewrote the entire word and his acronym in () first time it is reused in discussion. like l 536-38 be consistent with l 581”: in the revised manuscript, the acronyms were explained when they first appeared in every new section of the discussion.
“It is the first study with polysomnography onboard and it reveals impact of lenght of embarkment, age. It is important for sleeping and cardiovascular diseases prevention. It adds new knowledge on sleeping architecture, the feasibility of technical assessment of sleeping on board ships. Seafarers have higher risks of obstructive sleep apnea.”: we included a new paragraph to open our Discussion which further emphasizes these statements.
“Sometimes the Discussion is not easy to follow with a lot of abbreviations and some parts are very technical”: since our study generally deals with many different parameters and measurement methods, it was unfortunately not possible for us to use less technical descriptions without losing valuable information for future studies. However, we have re-explained the abbreviations at the beginning of each new discussion section so that a better understanding of the text is possible. In addition, we have made minor changes (e.g., lines 402-403) to allow a better reading flow.
Reviewer 2 Report
It is appropriate that the authors define the study as a pilot study due to the low response rate.
The introduction is very crammed, and it is suggested that the 'as well as loneliness - as a result of an isolated environment' is taken out. It is not referred to further.
The authors might consider pointing out how evidence has been found to support the argument that there is a connection between fatigue and errors, accidents, and injury (Techera et al. 2016; Williamson et al. 2011; Robb et al. 2008).
They may also consider including that for some health issues, latency time can be many years later, and at great cost. Alone in the US, costs of lost productive time in fatigued workers, has been estimated at 101 billion USD per annum compared to non-fatigued workers (Ricci et al. 2007). These consequences can result in early retirement for some workers.
The maritime terminology could be thought through in order to assist the readers. It is not certain that non-maritime will even know what a watchkeeper is or a deck rating for that matter. Maybe just use navigation officers and use it it consistently throughout the paper. Consider something for ratings or a footnote to explain. Be consistent and use seafarers throughout, not sailors -page 2, 2nd paragraph, 2nd word. Or do the authors refer to the yachting sector here?
It is suggested that the authors take out the yacht racing text - 4 sentences - on page 2, it is not comparable.
On page 2, parag. 5 'A polysomnographic ....' include 'onboard' or 'at sea'.
Good to include Figure 1, which assists the reader in understanding.
Section 3.results; begins with, 'This section may be divided....' these three sentences read as assistance to authors on how to write the results section. Suggest to rephrase, also as the authors have decided to use subheadings.
The discussion section begins the same way as the Results section. it provides an odd read. This discussion is very dense, suggest the authors thin it out where possible. It is important to underline again in the discussion and conclusion, that it is a pilot study and small, however the data suggests the importance of the subject and the necessity to undertake a larger study. This point could be made more clear here and in the abstract.
There are a few typos - please check.
The authors may consider these references;
Robb G, Sultana S, Ameratunga S, Jackson R. A systematic review of epidemiological studies investigating risk factors for work-related road traffic crashes and injuries. Inj Prev. 2008;14(1):51-58
Ricci JA, Chee E, Lorandeau AL, Berger J. Fatigue in the U.S. workforce: prevalence and implications for lost productive work time. J Occup Environ Med. 2007;49(1):1-10. 24.
Techera U, Hallowell M, Stambaugh N, Littlejohn R. Causes and consequences of occupational fatigue: meta-analysis and systems model. J Occup Environ Med. 2016;58(10):961-973.
Williamson A, Lombardi DA, Folkard S, Stutts J, Courtney TK, Connor JL. The link between fatigue and safety. Accid Anal Prev. 2011;43(2):498-515.
Dorrian, J. Hussey, F, Dawson D. Train driving efficiency and safety: examining the cost of fatigue. Journal of Sleep Res. 2007; 16 (1): 1-11. Doi: 10.1111/j.1365-2869.2007.00563.x
It seems odd that the World Maritime University report (2020) is not included, "A culture of adjustment. Evaluating the implementation of the current maritime regulatory framework on rest and work hours". This report was discussed both at the IMO and at a MLC 2006 tripartite meeting.
Author Response
We would like to thank the reviewer for the helpful comments that are valuable for improving the quality of the present manuscript. In this response, the reviewers’ comments are presented in bold italic letters and followed by our answers:
Reviewer 2
“The introduction is very crammed, and it is suggested that the 'as well as loneliness - as a result of an isolated environment' is taken out. It is not referred to further.”: in the revised version we deleted this suggested part for a better reading flow.
“The authors might consider pointing out how evidence has been found to support the argument that there is a connection between fatigue and errors, accidents, and injury (Techera et al. 2016; Williamson et al. 2011; Robb et al. 2008).”: we added these references (lines 104-105) to further support associations of fatigue and errors, accidents, and injury in different studies.
“They may also consider including that for some health issues, latency time can be many years later, and at great cost. Alone in the US, costs of lost productive time in fatigued workers, has been estimated at 101 billion USD per annum compared to non-fatigued workers (Ricci et al. 2007). These consequences can result in early retirement for some workers.”: this is indeed an intriguing issue. However, the main focus of our study is on the feasibility of the measurements on board, as well as on the current condition of the seafarers. Therefore, we have decided not to examine these long-term effects in more detail to keep the length of the publication within reasonable limits.
“The maritime terminology could be thought through in order to assist the readers. It is not certain that non-maritime will even know what a watchkeeper is or a deck rating for that matter. Maybe just use navigation officers and use it consistently throughout the paper. Consider something for ratings or a footnote to explain. Be consistent and use seafarers throughout, not sailors -page 2, 2nd paragraph, 2nd word. Or do the authors refer to the yachting sector here?”: in the revised manuscript we explained the terms “officer, non-officer (instead of the term “ratings”), watchkeeper, and day worker” in more detail in the Methods. Moreover, we changed the expressions “sailors” to “seafarers” as well as “deck ratings” to “crew ratings on deck”.
“It is suggested that the authors take out the yacht racing text - 4 sentences - on page 2, it is not comparable.”: this paragraph has been erased.
“On page 2, parag. 5 'A polysomnographic ....' include 'onboard' or 'at sea'.”: the addition “on board” has been inserted.
“Section 3.results; begins with, 'This section may be divided....' these three sentences read as assistance to authors on how to write the results section. Suggest to rephrase, also as the authors have decided to use subheadings.”: this paragraph has been deleted.
“The discussion section begins the same way as the Results section. it provides an odd read. This discussion is very dense, suggest the authors thin it out where possible. It is important to underline again in the discussion and conclusion, that it is a pilot study and small, however the data suggests the importance of the subject and the necessity to undertake a larger study. This point could be made more clear here and in the abstract.”: the first paragraph of the discussion has been deleted. We have implemented some minor changes in the discussion to improve the reading flow. However, we were unable to delete larger sections without losing valuable information for future studies. The recommendation to emphasize the pilot study design and the importance of the results has been included in the conclusion (lines 697-701).
“There are a few typos - please check.”: this comment has been implemented.
“The authors may consider these references;
Robb G, Sultana S, Ameratunga S, Jackson R. A systematic review of epidemiological studies investigating risk factors for work-related road traffic crashes and injuries. Inj Prev. 2008;14(1):51-58
Ricci JA, Chee E, Lorandeau AL, Berger J. Fatigue in the U.S. workforce: prevalence and implications for lost productive work time. J Occup Environ Med. 2007;49(1):1-10. 24.
Techera U, Hallowell M, Stambaugh N, Littlejohn R. Causes and consequences of occupational fatigue: meta-analysis and systems model. J Occup Environ Med. 2016;58(10):961-973.
Williamson A, Lombardi DA, Folkard S, Stutts J, Courtney TK, Connor JL. The link between fatigue and safety. Accid Anal Prev. 2011;43(2):498-515.
Dorrian, J. Hussey, F, Dawson D. Train driving efficiency and safety: examining the cost of fatigue. Journal of Sleep Res. 2007; 16 (1): 1-11. Doi: 10.1111/j.1365-2869.2007.00563.x
It seems odd that the World Maritime University report (2020) is not included, "A culture of adjustment. Evaluating the implementation of the current maritime regulatory framework on rest and work hours". This report was discussed both at the IMO and at a MLC 2006 tripartite meeting.”: the references Techera et al. 2016 [17]; Williamson et al. 2011 [18] and Robb et al. 2008 [16] were included in the introduction as proposed (see above). Additionally, we added the findings of the World Maritime University report (2020) in lines 56-59.
